# Body Composition and Determination of Somatotype of the Spanish Elite Female Futsal Players

Mónica Castillo [1], Isabel Sospedra [1,2,*], Estela González-Rodríguez [1], José Antonio Hurtado-Sánchez [1], Mar Lozano-Casanova [1], Rubén Jiménez-Alfageme [1] and José Miguel Martínez-Sanz [1,2]

1. Nursing Department, Faculty of Health Sciences, University of Alicante, 03690 Alicante, Spain; monica.castillo@ua.es (M.C.); estela.gonzalez@ua.es (E.G.-R.); ja.hurtado@ua.es (J.A.H.-S.); mlc50@gcloud.ua.es (M.L.-C.); rja10@gcloud.ua.es (R.J.-A.); josemiguel.ms@ua.es (J.M.M.-S.)
2. Ibero-American Network of Researchers in Applied Anthropometry, 04120 Almeria, Spain
* Correspondence: isospedra@ua.es

**Featured Application:** This research shows the results of the body composition of Spanish female futsal athletes, but until now, no such sample had been analysed in Spain. These findings could be a guide for coaches and physical trainers to focus on and individualise training objectives to achieve maximum performance, in addition to the fact that these data could be useful for the detection of talent for a specific position in young players.

**Abstract:** Background: In any sport, it is of the utmost importance to identify variables capable of positively influencing scores. Futsal is a sport of intermittent high-intensity intervals that requires the athletes' physical, technical, and tactical skills. There are no specific recommendations on anthropometry–body composition for this sport. The aim of this study was to describe the somatotype and the anthropometric characteristics, body composition, and somatotype of a group of Spanish elite female futsal players of the first-division league. Methods: Eighty-eight players (24 ± 4.94 years) from 14 teams of the first women's division of the Spanish Futsal League were evaluated. Measurements were taken according to the anthropometric protocol of the International Society for the Advancement of Kinanthropometry (ISAK). Body composition (BC) and somatotype were calculated according to the 4-component model and Heath-Carter, respectively. Results: Pivot and goalkeeper positions were most different from the rest. Both were the heaviest positions and presented the largest mesomorphic data (4.13 ± 1.29 and 3.67 ± 0.95), kg of bone mass (10.60 ± 1.00 and 10.37 ± 1.08), and kg of lean mass (29.80 ± 1.25 and 29.12 ± 2.12), for the pivot and goalkeeper, respectively. Conclusion: The evaluation and evolution of the somatotype and BC is an assessment tool that could be a useful guide for technical and medical staff.

**Keywords:** anthropometry; indoor football; tactical position; futsal performance; elite athletes





## 1. Introduction

Female futsal is among the growing sports worldwide, having more than 12 million players spread over 100 countries [1].

However, there is less information about the physical demands on female futsal players than male futsal players in the literature. Alvarez et al. [2] concluded that futsal is a multiple-sprint sport, in which there are higher-intensity phases than in soccer or other intermittent sports. During a futsal match, approximately 80% of the time is spent at an intensity above 85% the maximum heart rate (MHR). This sport shares physical characteristics, such as explosive movements, repetitive sprints, high demand for VO$_2$, and repeated plyometric requirements, and requires anaerobic and aerobic metabolic pathways to obtain energy [2].

These physical demands have been related to body composition and somatotype in different sports [3–6]. Collins et al. [7] concluded that higher levels of fat mass (FM) on

athletes could have detrimental effects on performance elements, such as speed, agility, and endurance. In contrast, an increase in lean mass (LM) could improve endurance and strength [8]. In addition, the risk of injury in this sport and the possible influence of anthropometric characteristics (such as weight and body composition) must be considered. It is important to highlight that the risk of injury in futsal is higher than football [9,10]; however, little is known about how anthropometric characteristics may impact injury risk [10].

Anthropometric measures, BC, and somatotype variables vary among sports, due to the selection criteria, hours of training, and sports-specific demands during the game and practice. This is one of the reasons why body composition assessment is important, as this knowledge could be a determining factor in sport performance [11–13].

In sports practice, it is essential to identify the main variables that can positively influence the match's outcome. BC must be assessed to perform training planning. This is due to BC being related to the improvement of many skills for high-level competition, including movement techniques, specific physical fitness, agility, skills, and performance [12]. Overall, the percentage of FM and lean mass (LM) are determinative factors of BC related to training adaptations results [13]. Furthermore, somatotype representation is also used in the assessment of BC related to sports performance, and it has been associated with successful results within sports. The Heath–Carter method [14,15] is the most common method used to assess somatotype. It gives us a holistic quantification of the morphology and characteristics of an athlete's body, using measures related to body shape and composition. The somatotype is defined as the quantification of the current shape and position of the human body. This is based on a three-number rating system, corresponding to different compartments of BC: relative adiposity (endomorph), muscle-skeletal robustness (mesomorph), and linearity or slenderness. The anthropometric technique is used to obtain BC, the proportionality body index (ectomorph), and somatotype. The somatotype chart is a visual way to compare the anthropometric characteristics of the studied athletes versus other players with successful results, or with elite players in a specific sport [11]. Currently, there are few studies on body composition in female futsal players. A few of them include a description of the somatotype; in others, the samples are not representative [12,13,16,17].

The aim of this study was to describe the anthropometric characteristics, body composition, and somatotype of a group of Spanish elite female futsal players of the first-division league. As in other team sports, where the body composition changes according to the positions, it could also occur in female futsal, where a specific somatotype and body composition could be key factors to reach a high profile in the selected area of the sport. The results could be of use as guides and markers.

## 2. Materials and Methods

### 2.1. Subjects

The study population was selected through non-probabilistic convenience sampling. Eighty-eight Spanish elite female futsal players belonging to 14 different teams of the Spanish futsal 1st division participated voluntarily in this study. The assessment was carried out from October 2019 to March 2020, corresponding to the competitive season. A written consent form was given to each participant to be previously informed of the objective and characteristics of the study.

The ethics committee of the University of Alicante approved this research with file number UA-2018-05-22. Each participant was previously informed about the objective and characteristics of the study by an informed consent form designed following the Helsinki Declaration guidelines of the World Medical Association for medical research in human subjects.

### 2.2. Experimental Design

This was an observational and descriptive study. The population studied included Spanish elite female futsal players, and the exclusion criteria were: (1) players competing in

leagues other than the Spanish one; (2) players of divisions different from the 1st division; and (3) players with any active injuries.

### 2.3. Data Collection Procedure

Anthropometric measurements were taken according to ISO 7250-1:2017 and the International Society for the Advancement of Kinanthropometry (ISAK) standard [18].

The equipment used for the anthropometric measurements consisted of a standing height rod (1 mm accuracy), a weight scale (100 g accuracy, model TANITA-BC-601, Amsterdam, The Netherlands), a flexible measuring tape (1 mm accuracy), a small sliding calliper (1 mm accuracy, Calibres Argentinos SRL, Rosario, Argentina), a skinfold calliper (0.2 mm accuracy, Holtain, Crymych, UK), and other supplementary tools (a demographic pencil and an anthropometric box).

A trained anthropometrist with level III ISAK certification conducted the assessment. The BC measures included: (1) basic measures (weight and height); (2) skinfolds (biceps, triceps, subscapular, iliac crest, supraspinal, front thigh, medial calf); (3) girth (arm relaxed, arm flexed and tensed, wrist, gluteal, calf); and (4) breadths (humerus, femur, biacromial). BC was determined by formulas described in BC assessment in sports medicine of the Spanish Group of Kinanthropometry (GREC) [19], following the four-component model (muscular mass (MM), FM, bone mass (BM) and residual mass (RM)). The following formulas were used: (1) Wither's formula to calculate FM expressed in percentage; (2) Lee's formula to calculate MM expressed in kg; and (3) Rocha's formula to calculate BM expressed in kg. At the same time, the sum of 8 skinfolds was calculated, as well as two health indexes: the waist-height index and the body adiposity index. Technical measuring errors were 7.5% for the skinfolds and 1.5% for the rest of the measures [20].

Somatotype was estimated following the Heath–Carter method, establishing the three Carter components (endomorph, mesomorph, and ectomorph, separately) and representing those results in a somatotype chart. The somatotype chart is the graphical representation of the somatotype where the rating of the three components of the somatotype is plotted in a two-dimensional chart [21].

### 2.4. Statistical Analysis

SPSS v23.1 (IBM España S.A, Madrid, Spain) software was selected to perform the statistical analysis.

The Kolmogorov–Smirnov test was used to check the normal distribution of data. Spearman's rank correlation test was used to assess no normal distributed data (age, bicipital skinfold, front thigh girth, bystiloid girth, femur girth, biepicondylar girth, and bone mass percentage), and Pearson's correlation coefficient test was used to assess the correlation between normally distributed data. A paired sample *t*-test and Mann–Whitney–Wilcoxon test were used to determine differences between the data of different tactical positions. *p* values below 0.05 were considered statistically significant, and data were presented as means and standard deviations (X ± SD). The tactical positions were established based on the categorisation of the Spanish women's futsal team: Goalkeeper (G), Forward (F), Wing (W), Wing-Pivot (WP), and Pivot (P) [22].

## 3. Results

To establish the characterisation of each tactical position existing in futsal, the results are shown in tables organised by tactical position and the total average of the sample.

Table 1 shows the basic data used to calculate BC, as well as weight, height, girth, perimeter, and skinfolds.

P and G positions were the most similar positions, as no significant differences were found between them. At the same time, positions P and G showed significant differences with respect to the other positions. P and G were the heaviest positions, presenting significant differences with F, WP, and W. The G group was the tallest with a height of 1.70 ± 0.06 m, a value that was significantly different from F, WP, and W.

**Table 1.** Anthropometric characteristics of female elite futsal players according to tactical position.

| | | Total (*n* = 88) | Wing-Pivot (*n* = 28) | | Pivot (*n* = 7) | | Wing (*n* = 17) | | Forward (*n* = 20) | | Goalkeeper (*n* = 16) | |
|---|---|---|---|---|---|---|---|---|---|---|---|---|
| | | Mean ± SD | Mean ± SD | * *p* Value | Mean ± SD | * *p* Value | Mean ± SD | * *p* Value | Mean ± SD | * *p* Value | Mean ± SD | * *p* Value |
| **Basic measures** | Age (years) | 24.00 ± 4.94 | 22.71 ± 3.59 | | 25.00 ± 4.93 | | 23.25 ± 4.09 | | 26.89 ± 6.35 | | 24.00 ± 5.71 | |
| | Weight (kg) | 59.82 ± 6.76 | 60.36 ± 3.55 | | **65.43 ± 6.51 *** | F *p* = 0.029 W *p* = 0.005 WP *p* = 0.004 | 55.65 ± 4.77 | | 58.63 ± 7.69 | | **64.31 ± 7.47 *** | F *p* = 0.016 W *p* = 0.003 WP *p* = 0.004 |
| | Height (m) | 1.65 ± 0.06 | 1.68 ± 0.04 | | 1.66 ± 0.06 | | 1.62 ± 0.05 | | 1.63 ± 0.06 | | **1.70 ± 0.06 *** | F *p* = 0.001 W *p* = 0.001 WP *p* = 0.004 |
| **Breadths** | Bystiloid (cm) | 6.22 ± 0.45 | **6.24 ± 0.51 *** | W *p* = 0.015 | **6.51 ± 0.34 *** | F *p* = 0.003 W *p* = 0.003 WP *p* = 0.035 | 5.99 ± 0.39 | | 6.27 ± 0.35 | | **6.38 ± 0.59 *** | W *p* = 0.003 |
| | Femur (cm) | 9.04 ± 0.65 | 9.05 ± 0.89 | | **9.44 ± 0.39 *** | W *p* = 0.013 WP *p* = 0.031 | 8.73 ± 0.37 | | 9.09 ± 0.53 | | **9.19 ± 0.25 *** | W *p* = 0.004 |
| | Humerus (cm) | 5.19 ± 0.35 | 5.27 ± 0.33 | | **5.57 ± 0.57 *** | W *p* = 0.013 WP *p* = 0.035 | 4.92 ± 0.28 | | **5.09 ± 0.35 *** | W *p* = 0.045 | 5.29 ± 0.53 | |
| **Girths** | Relax arm (cm) | 26.40 ± 1.89 | 25.79 ± 1.68 | | **27.63 ± 1.12 *** | W *p* = 0.036 WP *p* = 0.020 | 25.38 ± 2.21 | | 26.39 ± 1.56 | | 27.11 ± 2.48 | |
| | Flexed and tensed arm (cm) | 27.51 ± 1.74 | 27.08 ± 1.58 | | **28.54 ± 1.16 *** | WP *p* = 0.012 | 26.83 ± 2.02 | | 27.74 ± 1.52 | | **28.16 ± 2.39 *** | WP *p* = 0.049 |
| | Front thigh (cm) | 52.41 ± 2.97 | 52.21 ± 2.78 | | **54.69 ± 3.03 *** | F *p* = 0.023 W *p* = 0.033 WP *p* = 0.039 | 51.54 ± 2.86 | | 51.61 ± 2.63 | | **53.33 ± 3.57 *** | F *p* = 0.036 W *p* = 0.015 |
| **Girths** | Calf (cm) | 35.25 ± 2.85 | 34.76 ± 4.45 | | 37.70 ± 2.15 | F *p* = 0.001 W *p* = 0.013 WP *p* = 0.004 | 34.49 ± 1.45 | | 34.93 ± 2.02 | | 36.43 ± 2.27 | F *p* = 0.029 W *p* = 0.005 |
| | Waist (cm) | 71.89 ± 4.04 | 72.47 ± 3.37 | | **74.36 ± 3.31 *** | W *p* = 0.016 | 68.95 ± 2.94 | | 71.46 ± 5.09 | | 72.75 ± 4.00 | |
| | Gluteal (cm) | 95.88 ± 5.18 | 94.85 ± 3.98 | | **100.19 ± 5.96 *** | WP *p* = 0.006 | 94.88 ± 5.16 | | 94.83 ± 6.27 | | 97.39 ± 5.57 | |

**Table 1.** *Cont.*

| | | Total (*n* = 88) | Wing-Pivot (*n* = 28) | | Pivot (*n* = 7) | | Wing (*n* = 17) | | Forward (*n* = 20) | | Goalkeeper (*n* = 16) | |
|---|---|---|---|---|---|---|---|---|---|---|---|---|
| | | **Mean ± SD** | **Mean ± SD** | **\* *p* Value** | **Mean ± SD** | **\* *p* Value** | **Mean ± SD** | **\* *p* Value** | **Mean ± SD** | **\* *p* Value** | **Mean ± SD** | **\* *p* Value** |
| **Skinfolds** | Subscapular (mm) | 10.46 ± 2.97 | 10.11 ± 2.87 | | **12.94 ± 4.59 \*** | W *p* = 0.029 | 9.86 ± 1.84 | | 10.36 ± 2.66 | | 11.19 ± 2.44 | |
| | Tricipital (mm) | 14.16 ± 3.89 | 13.96 ± 3.57 | | 15.64 ± 5.73 | | 14.90 ± 4.57 | | 13.06 ± 3.31 | | **15.39 ± 2.85 \*** | F *p* = 0.015 |
| | Bicipital (mm) | 5.69 ± 1.99 | **6.19 ± 1.99 \*** | W *p* = 0.025 | **7.76 ± 1.77 \*** | F *p* = 0.042 W *p* = 0.004 | 4.90 ± 1.37 | | 5.44 ± 2.20 | | 5.46 ± 2.09 | |
| | Iliac crest (mm) | 14.44 ± 4.58 | 15.84 ± 3.75 | | 15.84 ± 6.39 | | 12.84 ± 4.59 | | 14.81 ± 5.15 | | 14.89 ± 4.72 | |
| | Supraespinale (mm) | 14.67 ± 4.49 | 14.36 ± 4.29 | | **18.00 ± 4.89 \*** | F *p* = 0.049 | 12.38 ± 4.23 | | 13.72 ± 3.60 | | 17.10 ± 4.72 | |
| | Abdominal (mm) | 15.33 ± 5.78 | 15.78 ± 5.25 | | 18.36 ± 8.74 | | 15.50 ± 5.46 | | 15.48 ± 5.32 | | 17.04 ± 6.04 | |
| | Front thigh (mm) | 21.87 ± 5.75 | 22.45 ± 4.99 | | 22.11 ± 8.12 | | **25.75 ± 6.87 \*** | F *p* = 0.045 | 20.42 ± 5.46 | | 21.73 ± 4.73 | |
| | Medial calf (mm) | 10.92 ± 3.77 | 11.43 ± 3.63 | | 10.86 ± 3.92 | | 12.11 ± 3.97 | | 9.80 ± 3.78 | | 11.77 ± 4.11 | |
| | 6 skinfold sum (mm) | 82.30 ± 23.18 | 70.86 ± 19.52 | | 92.76 ± 37.61 | | 82.31 ± 21.84 | | 71.97 ± 17.36 | | 88.87 ± 21.93 | |
| | 8 skinfold sum (mm) | 101.51 ± 29.15 | 109.48 ± 23.34 | | 114.15 ± 27.63 | | 98.80 ± 26.52 | | 95.77 ± 27.77 | | 107.52 ± 28.37 | |

\* *p* value: Significant *p* value (*p* < 0.05). Abbreviations: F = Forward; G = Goalkeeper, P = Pivot, W = Wing, WP = Wing-Pivot.

Bone girth followed the same line as the basic anthropometric measurements for P and G positions when compared to the rest. P position had significant differences in bystiloid girth compared to positions F, WP, and W. Bystiloid girth also showed differences between G and W. Femur girth presented significant differences between P and WP and W, as well as between G and W.

Regarding the breadth measures, P presented a larger relax arm breadth than W and WP. Calf and front tight breadths of P were also significantly larger compared to W, WP, and F. Concerning calf girth, G position presented larger measures than F and W. WP showed the lowest breadth of flexed and tensed arm compared with G and P. Finally, P was the position with the highest waist and gluteal breadth, presenting significant differences with W position in waist breadth and in gluteal breadth with WP position.

Regarding the skinfolds, the P position again showed the most differences with the other groups. Significant differences were mainly observed when P was compared to W and F. P presented larger values in subscapular and bicipital compared to W. When comparing P to F, larger values were observed in P for bicipital and supraspinal skinfolds. As for the tight skinfold, the only difference found was between the W and F positions.

Table 2 contains the description of three categories of BC recorded (muscle mass, bone mass, and fat mass) and some related indexes.

P and G had the highest MM ($29.80 \pm 1.25$ and $29.12 \pm 2.12$ kg, respectively), and these values were different from F, WP, and W. BM (kg) for P and G were higher than F, W, and WP. Regarding the FM measurement, G and P were not significantly different, and G (but not P) was significantly different from W, WP, and F. Concerning the free fat mass index (FFMI) (kg/m$^2$), P was higher than G and WP. Active body mass was higher for the P and G positions, and both were significantly different from W and WP. Additionally, P was different from F. The G position had the lowest body adiposity index and was significantly different from P, F, and W. Finally, the mesomorphic component was higher in P compared to W and WP. For this same measurement, F was higher than WP.

Figure 1 is the representation of the somatotype of each position. All positions were represented as endo-mesomorph somatotypes, despite possible significant differences between some anthropometric data.

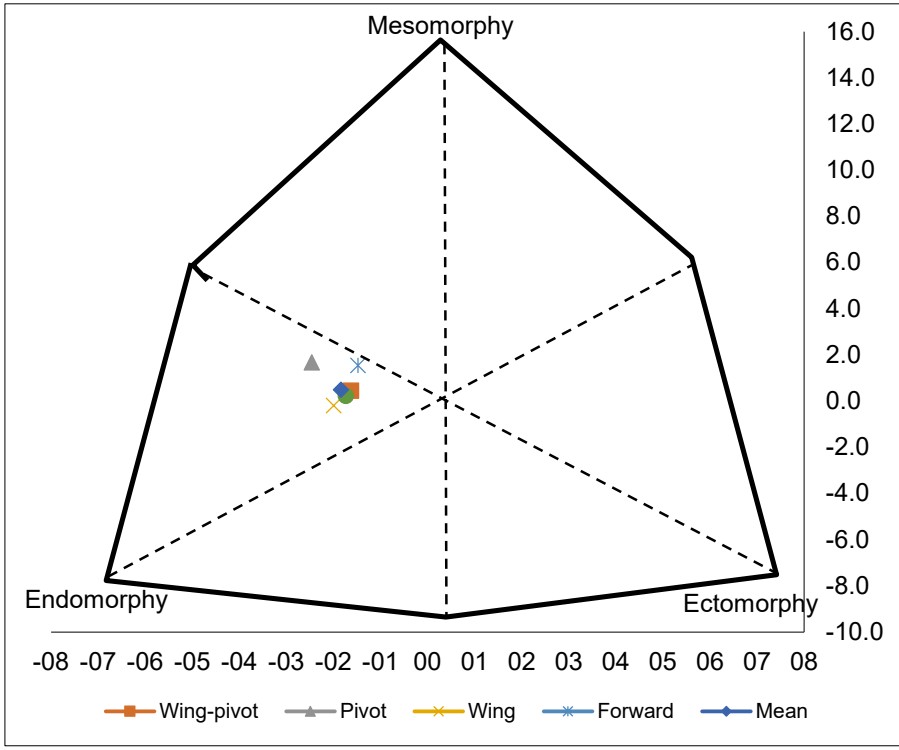

**Figure 1.** Representation of the somatotype.

**Table 2.** Average and deviation of BC values, index, and somatotype values, and classification according to tactical position.

| | | Total (*n* = 86) | Wing-Pivot (*n* = 28) | | Pivot (*n* = 7) | | Wing (*n* = 17) | | Forward (*n* = 20) | | Goalkeeper (*n* = 14) | |
|---|---|---|---|---|---|---|---|---|---|---|---|---|
| | | Mean ± SD | Mean ± SD | * *p* Value | Mean ± SD | * *p* Value | Mean ± SD | * *p* Value | Mean ± SD | * *p* Value | Mean ± SD | * *p* Value |
| **Body composition** | MM Lee (kg) | 27.89 ± 2.04 | 27.51 ± 1.30 | | **29.80 ± 1.25 *** | F *p* = 0.011 W *p* = 0.000 WP *p* = 0.008 | 26.07 ± 1.10 | | 27.60 ± 1.71 | | **29.12 ± 2.12 *** | F *p* = 0.048 W *p* = 0.001 WP *p* = 0.014 |
| | MM Lee (%) | 47.00 ± 4.41 | 46.03 ± 3.92 | | 46.04 ± 6.04 | | 47.08 ± 3.33 | | 47.59 ± 4.59 | | 45.55 ± 3.00 | |
| | BM Rocha (kg) | 9.68 ± 1.19 | 9.85 ± 1.30 | | **10.60 ± 1.00 *** | F *p* = 0.011 W *p* = 0.010 WP *p* = 0.018 | 8.80 ± 0.81 | | 9.41 ± 1.07 | | **10.37 ± 1.08 *** | F *p* = 0.07 W *p* = 0.004 |
| | BM Rocha (%) | 16.22 ± 1.42 | 16.42 ± 1.71 | | 16.24 ± 1.09 | | 15.84 ± 1.02 | | 16.12 ± 1.32 | | 16.20 ± 1.33 | |
| | FM Withers (kg) | 7.29 ± 3.19 | 7.71 ± 2.42 | | 8.93 ± 5.24 | | 7.07 ± 2.28 | | 6.83 ± 2.88 | | 8.91 ± 2.73 | |
| | FM Withers (%) | 11.93 ± 4.41 | 12.82 ± 3.76 | | 13.12 ± 7.18 | | 12.63 ± 3.57 | | 11.41 ± 3.32 | | 13.74 ± 3.37 | |
| | FM Faulkner (kg) | 8.34 ± 2.14 | 8.46 ± 1.45 | | 10.08 ± 3.31 | | 7.73 ± 1.58 | | 8.21 ± 2.14 | | 9.45 ± 2.02 | |
| | FM Faulkner (%) | 13.81 ± 2.43 | 14.10 ± 2.12 | | 15.13 ± 3.83 | | 13.84 ± 2.14 | | 13.87 ± 2.04 | | 14.65 ± 2.25 | |
| | FM Carter (kg) | 7.54 ± 2.41 | 7.12 ± 1.40 | | 8.35 ± 3.14 | | 6.77 ± 1.49 | | 6.78 ± 2.21 | | 7.89 ± 1.93 | |
| | FM Carter (%) | 11.56 ± 12.44 | 11.85 ± 1.97 | | 12.46 ± 3.74 | | 12.09 ± 2.04 | | 11.40 ± 2.32 | | 12.19 ± 2.16 | |
| | Average of Fat mass formulas (kg) | 7.54 ± 2.41 | 6.82 ± 1.71 | | 9.12 ± 3.87 | | 7.19 ± 1.74 | | 7.28 ± 2.38 | | **12.19 ± 2.16 *** | F *p* = 0.048 W *p* = 0.020 WP *p* = 0.020 |
| | Average of FM formulas (%) | 12.44 ± 3.01 | 12.92 ± 2.55 | | 13.57 ± 4.88 | | 12.85 ± 2.51 | | 12.23 ± 2.48 | | 13.53 ± 2.53 | |
| | Active Body mass | 52.53 ± 5.08 | 44.03 ± 3.10 | | **49.33 ± 5.36 *** | F *p* = 0.031 W *p* = 0.06 WP *p* = 0.004 | 41.95 ± 3.00 | | 43.84 ± 4.29 | | **48.40 ± 4.83 *** | W *p* = 0.016 WP *p* = 0.018 |

**Table 2.** *Cont.*

| | | Total (*n* = 86) | Wing-Pivot (*n* = 28) | | Pivot (*n* = 7) | | Wing (*n* = 17) | | Forward (*n* = 20) | | Goalkeeper (*n* = 14) | |
|---|---|---|---|---|---|---|---|---|---|---|---|---|
| | | Mean ± SD | Mean ± SD | * *p* Value | Mean ± SD | * *p* Value | Mean ± SD | * *p* Value | Mean ± SD | * *p* Value | Mean ± SD | * *p* Value |
| **Index** | Waist-heigh index | 0.44 ± 0.02 | 0.44 ± 0.02 | | 0.45 ± 0.02 | | 0.43 ± 0.02 | | 0.44 ± 0.03 | | 0.43 ± 0.02 | |
| | Body adiposity index | 27.25 ± 2.49 | 26.51 ± 2.29 | | **28.66 ± 1.92 *** | G *p* = 0.021 | 28.22 ± 2.51 | | 27.66 ± 2.60 | | **25.89 ± 2.71 *** | F *p* = 0.036 W *p* = 0.019 |
| | FFMI (kg/m$^2$) | 19.26 ± 8.49 | 19.04 ± 0.93 | | **20.44 ± 1.40 *** | G *p* = 0.041 W *p* = 0.041 WP *p* = 0.003 | 19.08 ± 1.38 | | 19.44 ± 1.61 | | 19.75 ± 1.40 | |
| **Somatotype** | Endomorph | 3.82 ± 1.09 | 4.01 ± 1.02 | | 4.28 ± 1.70 | | 3.96 ± 0.99 | | 3.78 ± 0.75 | | 4.15 ± 0.91 | |
| | Mesomorphic | 3.27 ± 1.07 | 3.17 ± 1.31 | | **4.13 ± 1.29 *** | W *p* = 0.046 WP *p* = 0.019 | 2.97 ± 0.92 | | **3.67 ± 0.95 *** | WP *p* = 0.026 | 3.06 ± 1.12 | |
| | Ectomorph | 2.01 ± 0.61 | 2.03 ± 0.50 | | 1.60 ± 0.70 | | 2.06 ± 0.57 | | 1.92 ± 0.66 | | 2.17 ± 0.61 | |
| | Classification | Endo-mesomorph | Endo-mesomorph | | Endo-mesomorph | | Endo-mesomorph | | Endo-mesomorph | | Endo-mesomorph | |

* Significant *p* value < 0.05; Abbreviations: F = Forward; G = Goalkeeper, P = Pivot, W = Wing, WP = Wing-Pivot, MM = muscle mass, BM = bone mass, FM: fat mass, RM = residual mass, FFMI = free fat mass index.



## 4. Discussion

The purpose of this study was to describe the anthropometric characteristics, body composition, and somatotype of elite women futsal players in the Spanish first division. The results indicate that G and P positions had a BC significantly different from the other tactical positions. Both positions had greater height and body weight compared to the other positions. This is also the case with the hip, thigh, and calf girths, in which both positions differed significantly from W, F, and WP. The MM average of all positions was 47.00% ± 4.41, which was higher than the mean average obtained in Alicia Canda's study of Spanish athletes of different modalities (40.4 ± 3.57) [23]. As for the FM, in Canda's study, the average calculated by Whiters' formula was 18.14% ± 5.07, higher than the one obtained by the same formula in our study group (12.44% ± 3.01).

The sports that most resemble futsal in terms of physical requirements are sports with aerobic and anaerobic demands. This includes sports, such as handball [24], rugby [25], basketball, and football [26]. It would be logical to think that the anthropometric results of these different modalities should be similar; however, each sport has its individual characteristics that make it unique. All of these sports share an endo-mesomorph somatotype; nevertheless, the BC and tactical position are important factors that determine the specific energy expenditure [27], capacity in sprints and aerobic/anaerobic performance and specific skills of the tactical position [28]. In rugby seven, there are no significant differences for BC; however, certain anthropometry characteristics, the height in this case, influence the tactical demarcation of players. This has been shown through defensive positions that have higher height averages than offensive positions [25]. This is one of the reasons why it could be important to know the anthropometric characteristics by position, as studied in other sports, with the goal of bettering the factors that can influence performance.

As for the estimation of fat and muscle percentages, there are more than 100 equations, most of them developed by multiple regressions, which allow the determination of FM% and MM from anthropometric variables. One of the main problems of error lies in the selection of the equation used to determine FM%. Since the athletes' samples have to be as homogeneous as possible, there is currently a problem with standardisation [29]. Hence, the use of the sum of several skinfolds as an individual adiposity index [21] has been proposed. Though the sum of 6 skinfolds (tricipital, subscapular, supraspinal, abdominal, thigh, and leg) is normally used, many authors also apply the sum of 8 skinfolds [30]. For the sake of simplicity, an increase in the sum of the skinfolds is an indication of an increase in FM, and vice versa. In this study, the sum of 8 skinfolds was 101.51 ± 29.15 mm, and the sum of 6 skinfolds was 82.30 ± 23.18 mm. Due to the lack of studies using the sum of 6 skinfolds in women, this study will compare the resulting data with Martinez-Riaza [31] in male hominids. The male mean value in Martinez-Riaza's [31] study (46.86 ± 8.50 mm) was much lower than the value for our female sample (82.30 ± 23.18 mm), showing a lower endomorphic component. In sports with similar characteristics, the value found for the sum of 6 skinfolds was also higher in females than males. These sports include rugby, whose elite female players have recorded values of 85 and 105 mm, basketball with values of 95 mm, and handball with values between 90 and 95 mm [23].

FFMI is an index that relates to LM and height. It should be noted that the mesomorphic component related to LM may have positive implications for the development of anaerobic performance, the main characteristic of the physical demands in futsal [6]. Higher FFMI values could give an advantage in performance since the greater the FFMI, the greater the muscle strength and power [32]. In our study, FFMI had a mean average of 19.26 kg/m$^2$ ± 8.49, higher than the sample mean of a study carried out by Canda [23] (17.29 kg/m$^2$ ± 1.38, for female players of a variety of sports), and similar to the data presented in the same study for female rugby players (18.5–19.5 kg/m$^2$) and female weightlifters (18.5–19.5 kg/m$^2$) [23]. In contrast, in female athletes with other physical requirements or sports with different energy demands, such as triathlon or velocity, the FFMI was lower (17.5–18.5 kg/m$^2$ and 16.5–17.5 kg/m$^2$, respectively) [23].

The differences found in somatotype and BC between tactical positions were similar to those obtained in studies referring to elite male players [31] and Brazilian elite female futsal players [12]. In the case of the somatotype, the positions that differed the most from the rest were the P and the G. Players in this position had higher endomorphic components, higher height and weight, and higher fat mass than the rest of the positions, according to previous studies [16,33,34].

The somatotype can influence the anaerobic capacities of athletes, and it is a tool that can be used to construct a specific training plan [6]. Futsal skills require, among others, high capacity of repeated sprints and anaerobic component [25], some studies found a correlation between repeated sprint capacity and somatotype components, overall mesomorphy, or greater LM or MM with higher power and strength, as well with better personal time in 100 m sprint [35,36], and major strength and power output would be related with mesomorphy and ectomorph components [37,38]. In our study, the pivot and goalkeepers had the highest scores for endomorph somatotype. Both positions need to expend less energy for sprints and jumps.

In our study, P and G positions presented the most significant differences compared to the other players. These differences may be due to the specific requirement of the tactical work of these positions, both in need of managing and occupying the space. However, in the study of Ramos-Campo [39], there were no significant differences between the different positions in 17 elite female futsal players.

Finally, it is necessary to emphasise that the study of body weight and composition can be useful for the prevention of injuries in female athletes [10]. Although some studies indicate that the anthropometric characteristics of women can represent an anatomical advantage against injuries [40,41], several studies reported general injury rates of 6.1 injuries/1000 h of exposure in female soccer players, [42] and a major incidence of medical assistance in elite male futsal players with higher endomesomorph and mesomorph components [31]. There is no scientific evidence of a relationship between anthropometry and injuries in elite female futsal players. More research is needed to determine if and to what extent anthropometric characteristics are related to the appearance of injuries [10].

One of the limitations of our study is the amount of the sample collected, despite the fact that it compiled data from more than 50% of the first-division futsal female players in Spain. Additionally, there is an insufficient number of studies carried out on the female futsal population. The majority of studies have focused predominantly on the male football population, which makes it difficult to compare with studies on female athletes or other sports. In addition, in this study, the P position is under-represented compared to the WP and W positions. P is a very specific position, and there are many teams that do not have the P position or have only one player of these characteristics. On the contrary, there usually are 3 or more players as W or WP positions. The skills of the W of WP positions are similar, and some coaches consider these positions interchangeable. These are skills directly related to body weight, amount of muscle mass, amount of fat mass, and ratio of muscle weight to bone mass.

## 5. Conclusions

Futsal is a sport with very explosive and fast movements, agility, and speed of reaction. These are qualities directly related to body weight, amount of muscle mass, amount of fat mass, ratio of muscle weight to bone mass, etc.

We can conclude that Spanish female futsal players are characterised by an endomesomorph somatotype, where the muscular and adipose component predominates, with a sum of 8 folds around 100 mm and a percentage of muscle mass of 47%. The most differentiated position from the rest is the pivot position, with a sum of 8 folds around 114 mm and 29.80 kg of muscle mass. These data could be related to the technical characteristics of the pivot position, which focus on the ability to hold the ball and use the body to protect it and turn towards the goal.

Therefore, establishing an appropriate body composition could be fundamental for achieving optimal performance. The information obtained could be useful to health and sports professionals for a better understanding of the association between anthropometry profile and tactical position. As such, knowing the ideal body composition for a particular sport could be extremely beneficial to determine the individual objectives and the best position for each athlete, aiming to optimise their performance.

**Author Contributions:** Conceptualisation, M.C., I.S., E.G.-R. and J.M.M.-S.; methodology, M.C., J.A.H.-S., M.L.-C. and R.J.-A.; software and formal analysis, M.C., M.L.-C. and R.J.-A.; investigation, M.C., I.S., E.G.-R., J.A.H.-S. and J.M.M.-S.; data curation, M.C., M.L.-C. and R.J.-A.; writing—original draft preparation, M.C., I.S., J.A.H.-S. and J.M.M.-S.; writing—review and editing, M.C., M.L.-C. and R.J.-A.; visualisation and supervision, I.S., E.G.-R., J.A.H.-S. and J.M.M.-S. All authors have read and agreed to the published version of the manuscript.

**Funding:** This research received no external funding.

**Institutional Review Board Statement:** This study was conducted according to the guidelines of the Declaration of Helsinki and approved by the ethics committee of the University of Alicante (ref.: UA-2018-05-22).

**Informed Consent Statement:** Informed consent was obtained from all subjects involved in the study to conduct and publish the study.

**Data Availability Statement:** The data presented in this study are available in the tables of this article. The data presented in this study are available on request from the corresponding author.

**Acknowledgments:** The authors thank all research subjects for their willingness and interest in participating. In addition, they would like to extend our gratitude to the coaches and trainers who helped and organise each measurement and Tara Taylor for her invaluable help in translating this paper into English.

**Conflicts of Interest:** The authors declare no conflict of interest.

## Abbreviations

Body Composition (BC), Lean Mass (LM), Muscle Mass (MM), Residual Mass (RM), Goalkeeper (G), Forward (F), Wing (W), Wing-Pivot (WP), Pivot (P), Fat Free Mass Index (FFMI).

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
