# Peer review of "Body Composition and Determination of Somatotype of the Spanish Elite Female Futsal Players"

_applsci, doi:10.3390/app12115708_

Round 1

Reviewer 1 Report

Dear authors, first of all, thank you for submitting your work.

In fact, apparently some formation errors (i.e., erased text, absence of the line number, and missing information in the tables) prevent you from being able to carry out a thorough review.

Therefore, I have not evaluated the manuscript and suggest that the document be resubmitted properly formatted and complete.
I will be happy to do so once the manuscript is in ideal condition for review.

Best Regards

Author Response

We attach a cover letter to the reviewer's comments.

Reviewer 2 Report

Dear authors,

I reviewed the research entitled ‘Body composition and determination of somatotype of the Spanish elite female futsal players'.
I have some comments as below. Please consider them for your decision.

1) I believe that somatotype might be valuable to gain high performance for athletes without injury. However, there were any investigations for their injury rates and performances. Therefore, your discussions were speculated based on some past researches mainly. Totally, this research design and its discussions were not sufficient but there's only a novelty which was targeted for elite female futsal athlete.

2) Who did judge the somatotype in the research? Please also explain for somatotype chart and its formula in the METHODS.

3) All values should be rounded off to the second decimal places in your results.

4) It is difficult to catch the result of table 1 and 2. Please modify them simply.

That's all. 

Author Response

(The authors gave the same response as above.)

Reviewer 3 Report

Review Comments

Castillo et al.: Body composition and determination of somatotype of the Spanish elite female futsal players

The authors present a paper aiming the description of the somatotype and the anthropometric characteristics of Spanish elite female futsal players. Among 88 futsal players of the First Spanish Division the Pivot- and Goalkeeper-position were found to be different from other player positions regarding somatotype and body composition. The authors point out, how important the somatotype and body composition may be for high performance in sport disciplines like futsal. The topic is interesting and important. However, there are a lot of possibilities to improve the quality of the paper.

Improvements are recommended in structuring the paper, especially in the introduction and results section.

Due to missing line numbers, it is difficult to make comments in a specific way.

General comments

At three parts of the paper the authors mention three different aims of the paper: Abstract: “…to describe the anthropometric profile”; Introduction: “to describe the somatotype and the anthropometric characteristics”; Discussion: “to characterise the BC”. As it is not clear to me throughout the whole paper, how the terms are used by the authors and how they are related, it makes it difficult to understand, what the aim is.

The structure of the paper is not well thought out. I would have found it helpful, to get more information which is presented in the discussion section in the introduction, already.

Specific comments

Introduction

As being a person not knowing anything about futsal, I would have found it helpful to get more information about the game and about the player positions, in particular. Furthermore, when searching for information about player-positions I found different terms in the www. It remains unclear to me, what a Wing-Pivot and a Forward is, as I could not find these terms in official descriptions neither in the literature the authors cited (No. 11).

There is missing a description of Heath-Carter-Method as well as of somatotypes. Please provide readers with more information.

For a better reading flow I would recommend to put paragraph 5 (starting with “anthropometric measures”) after paragraph 3.

Materials and Methods

  1. - 2.1. Subjects, line 4: assessment in which year? Please add.
  2. - 2.3 Data Collection Procedure: Please explain the abbreviation “MM”.
  3. Please explain the Heath-Carter Method (maybe in the introduction, at least in the methods section).
  4. Please define the somatotypes – maybe better in the introduction. Furthermore, connect this information with the relevance in other sports, as you started in the introduction.
  5. Statistical Analysis: reference 11 – I cannot find information about the player positions there. Please provide more information about the player positions and its (bodily) requirements in the introduction section.

Results

  1. The whole section should be improved regarding the structure. Results should be presented in order of main outcomes and position or measure.
  2. Tables. I have no idea how to improve, but layout of the tables should be changed to get a better overview.

Discussion

  1. Paragraph 2 is better placed in the introduction.
  2. I do not understand the following sentence: “These skills are not defined in futsal…” You defined them in based on your references.
  3. Please explain the abbreviation “FFMI” at the beginning of paragraph 4.
  4. In paragraph 7 you cite a study which did not show differences between different positions. How do you explain the heterogenous results?

Additional comment

The paper is submitted in the correction mode – there are crossed out lines and many, many needless blank spaces, as well as character errors.

Author Response

(The authors gave the same response as above.)

Reviewer 4 Report

This is an interesting topic that can provide useful information at the player selection stage. This is a sport that is not much advertised in the media.

Each sport has its own requirements, and, of course, the players must be in good physical condition to be able to train, contribute to the team result and win the game.

I kindly suggest the following details:

  1. In the abstract is written “Eighty-eight players (24±4.94 years) from seven teams”…. In section ….2.1 is written” elite female futsal players belonging to 14 different teams”. How is it in the end???? Maybe it can be clarified. The total number of subjects is 88.
  2. In section 2.3 is mentioned that “Anthropometric measurements were performed following the standardised guidelines of the International Society for the Advancement of Kinanthropometry (ISAK)”. It is important to analyse and mention that the basic protocol that defines the measurement of the different body measurements is given in the ISO 7250-1:2017/ Basic human body measurements for technological design - Part 1: Body measurement definitions and landmarks.

To insert an image explaining how the selected body measurements were taken (the measurements were taken according to the ISO 7250-1:2017 and ISAK standards.)

3. In section 2.3. is written,” BC was assessed with a height metre”. In anthropometry research (manual method), body height is measured with Martin anthropometer; it is incorrect to say BC. When the body is scanned, the values of the required dimensions are generated automatically (according to the specified list of dimensions).

To introduce a legend to explain the following terms (even it can be found at the bottom of the table)MM, FM, BM.

4. The individual values of the established parameters were analysed with SPSS software. The selection volume is very important for the accuracy of the results. Perhaps it is necessary to explain how the selection volume was determined and what the distribution within the volume is for each category of players: goalkeeper, pivot, wing, etc.

In order to make recommendations for different positions in the team based on the interpretation of anthropometric results, it is necessary to ensure the representativeness of the sample from which the data were taken (in order to transfer the conclusions from the sample to a population).

5. By interpreting the results obtained with SPSS and knowing the required performance and results of the best player at each position, it is possible to profile the player who is particularly well suited to a particular position in order to achieve the best possible results.

This would be an important help for the technical and medical staff.

This method can also be applied to other sports ( a new research direction). In this way, those who want to be part of a high-performance sports team can be evaluated to see if they are capable of doing what is needed and have all the necessary qualities to meet the demands of the position.

Author Response

(The authors gave the same response as above.)

Round 2

Reviewer 1 Report

Dear author, below you can find specific comments regarding your work:

line 40 - the authors report that there is little information about the physical demands of female futsal players, which is not true. There are numerous studies in the literature on the topic. Also because in the following paragraphs it presents information on this same topic, even on male and non-female futsal. This introductory paragraph should be reformulated with reference to the physical demands on female futsal players.

Line 47 - When mentioning that it has been widely studied, you should add supporting references

Lina 37 - 50 - In the initial phase of the article, it is essential to improve the link between paragraphs and ideas.

Line 80 - The study problem is not fully identified. It would be important to be able to cross-reference information from previous studies on the topic. It is essential to justify the need for a new study. This is because, contrary to what is mentioned, there are several studies in the literature that reported on anthropometric characteristics, body composition and somatotype

Line 87 - perhaps this information should be framed within topic 2.1

Line 88 - Why isn't a sample characterization done here?

Line 132 - (IBM, Madrid, Spain) this information is not correct, please rephrase

Line 150 - Why are some values missing from the table? Why aren't p values displayed for all variables? Some data are in bold, others are not in bold. Why is there no reference to the statistical data presented in the table in the description of the results? There are a number of limitations in the way the results are presented.

Line 207 - dear author, I believe that the discussion can be improved by creating associations specifically with futsal to the detriment of other modalities. Furthermore, with the broader review of the literature suggested above, you will certainly find more studies that can help to more effectively justify their results, which, as I mentioned earlier, are confusing.

Author Response

line 40 - the authors report that there is little information about the physical demands of female futsal players, which is not true. There are numerous studies in the literature on the topic. Also because in the following paragraphs it presents information on this same topic, even on male and non-female futsal. This introductory paragraph should be reformulated with reference to the physical demands on female futsal players.

Response of the authors: The paragraph has been reformulated according to reviewer suggestion.

Line 47 - When mentioning that it has been widely studied, you should add supporting references

Response of the authors: Following the reviewer's suggestions, bibliographical references have been added:

  • Cárdenas-Fernández, V., Chinchilla-Minguet, J. L., & Castillo-Rodríguez, A. (2019). Somatotype and Body Composition in Young Soccer Players According to the Playing Position and Sport Success. Journal of strength and conditioning research, 33(7), 1904–1911. https://doi.org/10.1519/JSC.0000000000002125
  • Lago-Peñas, C., Casais, L., Dellal, A., Rey, E., & Domínguez, E. (2011). Anthropometric and physiological characteristics of young soccer players according to their playing positions: relevance for competition success. Journal of strength and conditioning research, 25(12), 3358–3367. https://doi.org/10.1519/JSC.0b013e318216305d
  • Dellagrana, R. A., Guglielmo, L. G., Santos, B. V., Hernandez, S. G., da Silva, S. G., & de Campos, W. (2015). Physiological, anthropometric, strength, and muscle power characteristics correlates with running performance in young runners. Journal of strength and conditioning research, 29(6), 1584–1591. https://doi.org/10.1519/JSC.0000000000000784

Line 37 - 50 - In the initial phase of the article, it is essential to improve the link between paragraphs and ideas.

Response of the authors: The paragraphs have been reformulated for a better understanding between paragraphs and ideas.

Line 80 - The study problem is not fully identified. It would be important to be able to cross-reference information from previous studies on the topic. It is essential to justify the need for a new study. This is because, contrary to what is mentioned, there are several studies in the literature that reported on anthropometric characteristics, body composition and somatotype

Response of the authors: We appreciate the reviewer's comments to improve the manuscript, but studies that reported on anthropometric characteristics, body composition and somatotype in female futsal players have been indicated. The studies have been extracted from the main databases such as Pubmed or WOS.

The scientific literature on female futsal players is scarce, where most studies focus on male players. This aspect is one of the limitations of the study and has been commented on in the discussion section (last paragraph).

Line 87 - perhaps this information should be framed within topic 2.1

Response of the authors: This paragraph has been inserted in section 2.1.

Line 88 - Why isn't a sample characterization done here?

Response of the authors: The characteristaion of the sample is described in topic 2.1

Line 132 - (IBM, Madrid, Spain) this information is not correct, please rephrase

Response of the authors: The phrase has been reformulated propperly.

Line 150 - Why are some values missing from the table? Why aren't p values displayed for all variables? Some data are in bold, others are not in bold. Why is there no reference to the statistical data presented in the table in the description of the results? There are a number of limitations in the way the results are presented.e

Response of the authors: Those with significant p-values are marked in bold. Only significant data for each of the variables are indicated, following the suggestions of the editor and reviewers.

Line 207 - dear author, I believe that the discussion can be improved by creating associations specifically with futsal to the detriment of other modalities. Furthermore, with the broader review of the literature suggested above, you will certainly find more studies that can help to more effectively justify their results, which, as I mentioned earlier, are confusing.

Response of the authors: We appreciate the reviewer's comments to improve the manuscript, but studies that reported on anthropometric characteristics, body composition and somatotype in female futsal players have been indicated. The studies have been extracted from the main databases such as Pubmed or WOS.

The scientific literature on female futsal players is scarce, where most studies focus on male players. This aspect is one of the limitations of the study and has been commented on in the discussion section (last paragraph).

Reviewer 2 Report

Dear authors,

I have checked your revised manuscript. I have no further comment in your content. Unfortunately, I found error in writing large 'P' in table 2.

That's all.

Author Response

Large ‘P’ has been deleted because it was a typing error. p-value data are indicated in the corresponding boxes.

Reviewer 3 Report

Thank you for improving the paper. Please control some blanks and punctuation.

Author Response

Thank you for improving the paper. Please control some blanks and punctuation.

Response of the authors: the manuscript has been revised to correct blanks and punctuation.

Round 3

Reviewer 2 Report

Dear authors,

Again, I can find large 'P' in Table 2. It might be 'p value'. Please check it.

That's all.

Author Response

(The authors gave the same response as above.)
